# Reproducing the Past: A Dataset for Benchmarking Inscription Restoration

### Shipeng Zhu
School of Computer Science and Engineering, Southeast University
Key Laboratory of New Generation Artificial Intelligence Technology and Its Interdisciplinary Applications (Southeast University), Ministry of Education, China
Nanjing, China
shipengzhu@seu.edu.cn

### Hui Xue*
School of Computer Science and Engineering, Southeast University
Key Laboratory of New Generation Artificial Intelligence Technology and Its Interdisciplinary Applications (Southeast University), Ministry of Education, China
Nanjing, China
hxue@seu.edu.cn

### Na Nie
Nanjing University Museum, Nanjing University
The China Centre for Linguistic and Strategic Studies, Nanjing University
Nanjing, China
niena@nju.edu.cn

### Chenjie Zhu
School of Computer Science and Engineering, Southeast University
Key Laboratory of New Generation Artificial Intelligence Technology and Its Interdisciplinary Applications (Southeast University), Ministry of Education, China
Nanjing, China
chenjiezhu@seu.edu.cn

### Haiyue Liu
School of Computer Science and Engineering, Southeast University
Key Laboratory of New Generation Artificial Intelligence Technology and Its Interdisciplinary Applications (Southeast University), Ministry of Education, China
Nanjing, China
haiyueliu@seu.edu.cn

### Pengfei Fang
School of Computer Science and Engineering, Southeast University
Key Laboratory of New Generation Artificial Intelligence Technology and Its Interdisciplinary Applications (Southeast University), Ministry of Education, China
Nanjing, China
fangpengfei@seu.edu.cn

## ABSTRACT

Inscriptions on ancient steles, as carriers of culture, encapsulate the humanistic thoughts and aesthetic values of our ancestors. However, these relics often deteriorate due to environmental and human factors, resulting in significant information loss. Since the advent of inscription rubbing technology over a millennium ago, archaeologists and epigraphers have devoted immense effort to manually restoring these cultural imprints, endeavoring to unlock the storied past within each rubbing. This paper approaches this challenge as a multi-modal task, aiming to establish a novel benchmark for the inscription restoration from rubbings. In doing so, we construct the Chinese Inscription Rubbing Image (CIRI) dataset, which includes a wide variety of real inscription rubbing images characterized by diverse calligraphy styles, intricate character structures, and complex degradation forms. Furthermore, we develop a synthesis approach to generate "intact-degraded" paired data, mirroring real-world degradation faithfully. On top of the datasets, we propose a baseline framework that achieves visual consistency and textual integrity through global and local diffusion-based restoration processes and explicit incorporation of domain knowledge. Comprehensive evaluations confirm the effectiveness of our pipeline, demonstrating significant improvements in visual presentation and textual integrity. The project is available at: https://github.com/blackprotoss/CIRI.

## CCS CONCEPTS

• **Computing methodologies → Reconstruction**.

## KEYWORDS

Dataset Construction, Inscription Restoration, Diffusion Model, Chinese Ancient Culture

**ACM Reference Format:**
Shipeng Zhu, Hui Xue, Na Nie, Chenjie Zhu, Haiyue Liu, and Pengfei Fang. 2024. Reproducing the Past: A Dataset for Benchmarking Inscription Restoration. In *Proceedings of the 32nd ACM International Conference on Multimedia (MM '24), October 28–November 1, 2024, Melbourne, VIC, Australia.* ACM, New York, NY, USA, 10 pages. https://doi.org/10.1145/3664647.3680587

---

* Corresponding author: Hui Xue.

## 1 INTRODUCTION

Before the advent of printing, etching characters onto durable surfaces was a crucial method for preserving information [4]. Inscriptions shown in Figure 1, as a representation of this ancient technology, encapsulate various aspects of our ancestors. Over millennia, this cultural heritage has often withstood decay caused by time, environmental factors, and human activities. Originating from the Southern Dynasties (420 A.D. - 589 A.D.) in ancient China, rubbings serve as faithful records of original inscriptions without damaging the artifacts [35]. The method for producing rubbings involves

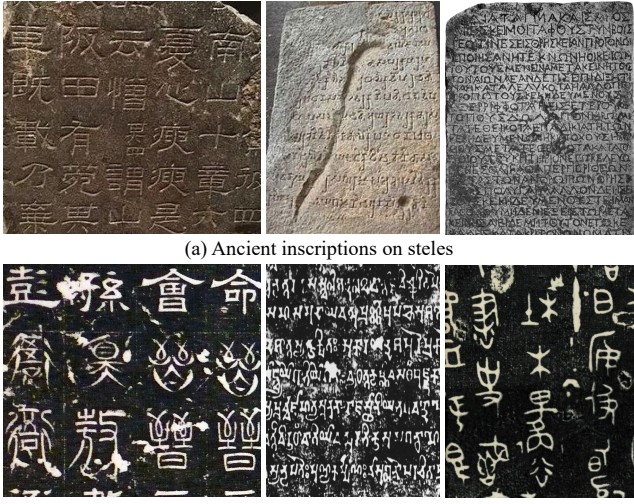

(a) Ancient inscriptions on steles

(b) Inscription rubbings on papers

**Figure 1: Examples of inscriptions and rubbings.**

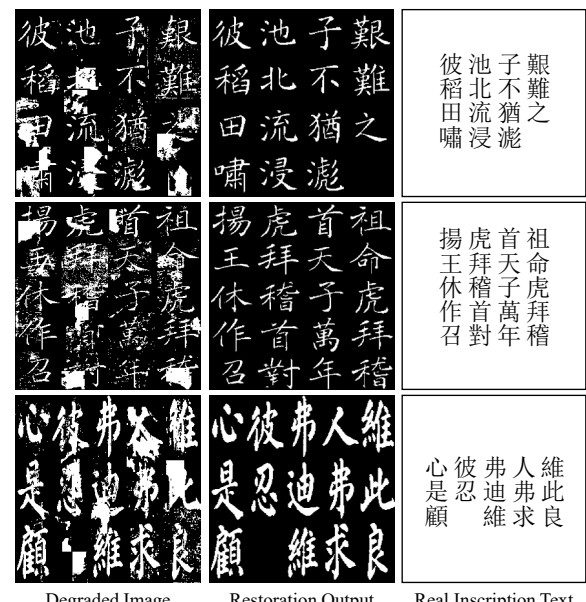

Degraded Image        Restoration Output        Real Inscription Text

**Figure 2: Restoration results of our pipeline on the CIRI.**

tightly covering the stone carving with a sheet of paper. Then, by applying black or colored materials over the paper, a detailed copy of the surface texture is obtained. Often, even when original steles are degraded or lost, rubbings offer a valuable glimpse into these inscriptions. Moreover, restoring inscriptions from these rubbings allows us to explore ancient calligraphic artistry, enrich historical narratives, and derive deep insights from classical texts [27, 36]. This propelled epigraphists to view rubbings as essential for preserving epigraphs [24], striving to reconstruct the consistent visual styles and logical textual content of the inscriptions.

Traditional inscription restoration has been deeply rooted in specialized expertise, drawing upon a profound understanding of historical literature and the unique stylistic nuances of various authors. Despite advancements in computer-aided technologies, the restoration process is still labor-intensive and time-consuming, primarily performed by dedicated scholars who carefully reconstruct the authentic content of these ancient artifacts.

With the advancement of artificial intelligence, it has become feasible to develop effective models for inscription restoration. In practice, researchers face two primary challenges: (1) **Unpaired Data**. Irreversible degradation obscures the original state of inscriptions, forcing researchers to use different rubbing images with varied levels of degradation. This reliance on unpaired data sources poses significant challenges in restoring inscriptions with high precision, both in text and style. (2) **Complex Degradation Form**. Long-term exposure to the environment subjects inscriptions to complex and unique degradations. For instance, defects such as scatter noise from rainwater and wind corrosion, along with large-scale peeling due to time or human activities, could lead to extensive character loss. These degradations create textual and visual ambiguities and challenge the accurate simulation of these patterns using computer graphics, complicating the construction of datasets for restoration.

Existing restoration models, trained on paired data, initially demonstrate excellent performance with natural images [11, 45]. In

scenarios with extensive missing areas, inpainting techniques are crucial for reconstructing logical content [48]. Nevertheless, these methods encounter substantial challenges with text-centric images like scene texts and documents, due to their inadaptability to the varied nature of text content [54]. Recently, tailored enhancement and restoration of documents have garnered considerable interest [33, 34, 42]. *While these approaches excel in achieving pixel-level fidelity, they often neglect the integrity of the restored textual content.* This oversight undermines the essence of ancient document studies, which relies on the faithful reconstruction and interpretation of the original texts. In epigraphy, pioneering efforts focus on the autonomous restoration of inscriptions. Among them, numerous studies focus on textual restoration [1, 2, 26], which aims to decipher lost languages from the remnants of characters. However, these efforts often fail to mimic the original visual presentation, such as the writing style—a crucial aspect of the information conveyed by inscriptions. Conversely, smaller initiatives that focus on visual restoration at the character level [38] face challenges with severely degraded inscriptions, where individual characters are often indistinguishable. *Concurrently, there is a notable gap in comprehensive studies on page-level inscriptions that closely reflect reality, crucial for restoring both visual consistency and textual integrity.*

This study utilizes the extensive collection of rubbing resources in China to develop the Chinese Inscription Rubbing Image (CIRI) dataset, establishing a new benchmark for evaluating inscription restoration methods. Initially, we collect 2000 rubbings from real epitaphs, spanning from the Han Dynasty (202 BC - 220 AD) to the Ming Dynasty (1368 AD - 1644 AD), and annotated them at the line level following [29]. Furthermore, to address the scarcity of paired rubbings, we develop an effective synthesis approach that integrates both real and simulated degradation patterns. This method produces rubbings with a range of degradation levels, effectively bridging

the gap in training data. As the essential extension to CIRI, we synthesize 240,000 "intact-degraded" paired images that closely mimic the real degraded inscriptions. With the CIRI dataset, we establish a robust foundation for the application of learning-based restoration techniques, enabling the precise restoration of visual presentation and textual content in ancient inscriptions.

Alongside the datasets, we introduce a specialized framework for inscription restoration, designed to emulate expert practices through the integration of domain knowledge. This pipeline begins with developing a diffusion-based Global Restoration Module (GRM), which effectively addresses easily identifiable defects in degraded images. Building upon this, we establish a Text Rectification Module (TRM) that draws directly from expert restoration techniques. This module initiates its process by predicting the overall layout and textual content of images restored by the GRM. Given the frequent occurrence of misspelled characters in recognition texts, which could potentially mislead the restoration with erroneous information, we employ text retrieval algorithms to extract text accurately from the real corpus. Furthermore, these verified segments are transformed into image format, thereby providing explicit guidance that reflects the correct structure of the corresponding ideographs. The next component, the Local Restoration Module (LRM), incorporates these image-format rectified texts as domain knowledge. Through character-by-character refinement, the LRM generates the final restoration output, achieving high visual fidelity and textual accuracy. Figure 2 vividly demonstrates the effectiveness and impact of our framework.

In a nutshell, our contributions are as follows:

- We propose a novel Chinese inscription restoration dataset, namely CIRI. As a pioneering work in complex inscription restoration, it comprises valuable existing rubbing images and realistic synthetic images, which provide a solid foundation for future restoration efforts.
- We introduce an effective pipeline for inscription restoration that aligns inherent style consistency with domain knowledge, achieving promising restoration outcomes.
- Extensive comparisons with existing methods demonstrate the superior performance of our method on both image quality and text recognition accuracy. Collectively, our benchmark dataset and tailored pipeline establish a robust platform for advancing future research in the social sciences.

## 2 RELATED WORK

### 2.1 Natual Image Restoration

The pursuit of restoring realistic and reasonable content from degraded images traditionally concentrates on natural scenarios [13]. Initial efforts have been narrowly focused on singular degradation scenarios, such as noise [5, 7] or blurring [50]. Of late, employing one universal framework for various degradation has emerged as a focal point of research [20, 45, 46]. Concurrently, alternative strategies regard the challenge as an image-to-image translation task [10, 25]. GDP [11] leverages the capabilities of diffusion models [16] for effective mapping, introducing hierarchical guidance and patch-based techniques to enhance restoration performance. Yet, these restoration methods focus on pixel-level precision and falter in addressing large-sized defects, a gap effectively bridged by

inpainting techniques [48]. Traditionally, leading inpainting strategies, such as [19, 23, 48], necessitate knowledge of the precise defect locations, which limit their applicability in diverse real-world contexts. Recent advancements in blind inpainting, such as [18, 39, 53], have moved beyond the dependence on predefined masks, predicting defect locations to guide model focus. Although these restoration methods demonstrate commendable performance for natural images, their application to text-centric images remains challenging. The inherent semantic sensitivity to text structure in such images poses unique challenges that are yet to be fully addressed.

### 2.2 Document Enhancement

Given the distinct pixel distribution differences between document images and their natural image counterparts [42], there is a pressing need for tailored document enhancement methods. Traditional approaches, as seen in [17, 22, 33, 34], focus on the separate modeling of foreground and background elements. Most recently, DocDiff [42] employs a diffusion-based series method that adeptly balances low- and high-frequency information in document images, showcasing notable efficacy across diverse degradation challenges. Yet, when it comes to the restoration of incomplete ancient texts [29], the majority of existing methods are limited to binary operations [41] or denoising [30, 49], leaving the complicated task of inferring missing text content predominantly to human researchers.

### 2.3 Text Reconstruction

Content loss due to degradation is a prevalent issue in various forms of texts, notably in ancient manuscripts. Traditional methods for reconstructing text, while effective, are notably time-consuming and labor-intensive [32]. In contrast, recent endeavors in autonomous text reconstruction, known as textual restoration [2, 24], have achieved satisfactory outcomes within the natural language processing community. A pioneering approach, Pythica [1] introduced a deep learning-based model for Greek inscription reconstruction at both word and character levels. This initiative attracts subsequent research applying deep-learning methods to diverse ancient texts, including Akkadian tablets [12], the Index Thomisticus Treebank [3], and Linear B inscriptions by Papavassileiou et al. [26]. Despite these advancements in textual restoration, restoring the visual presentation of inscriptions, which provides invaluable insights into ancient seal carving techniques and calligraphy art, remains a considerable challenge. To date, visual restoration efforts remain infrequent. Wang et al. [38] leverage Bert [8] for generating plausible glyphs for Chinese character inpainting, Although some contemporary studies [37, 54] address more complex defect scenarios, their scope is limited to word-level English inpainting. To sum up, comprehensive studies capable of tackling page-level inscription restoration with complex degradations remain absent.

## 3 DATASET CONSTRUCTION

### 3.1 Inscription Rubbing Image Collection

The goal of CIRI is to establish an extensive benchmark tailored for real-world scenarios in Chinese inscription restoration. To this end, we undertake considerable efforts to collect a broad spectrum of inscription rubbing images from electronic Stele collections available online. Concretely, our data collection strategy is guided by three

**Table 1: The degradation and attribute statistics of subsets in CIRI. Cls denotes the number of categories. The * denotes that the layout is categorized roughly based on the number of rows and columns. The - denotes not available.**

| Subset | Degradation | | | | Attribute | | | |
|---|---|---|---|---|---|---|---|---|
| | Easy | Medium | Hard | Total | Char/img | Char Cls | Font Cls | Layout Cls* |
| Training Set | 12,000 | 6,000 | 2,000 | 20,000 | 15.27 | 5264 | 186 | 9 |
| Testing Set S | 2,400 | 1,200 | 400 | 4,000 | 15.26 | 3901 | 186 | 9 |
| Testing Set R-I | 529 | 318 | 153 | 1000 | 13.20 | 879 | - | 9 |
| Testing Set R-II | 428 | 325 | 247 | 1000 | 13.40 | 703 | - | 9 |

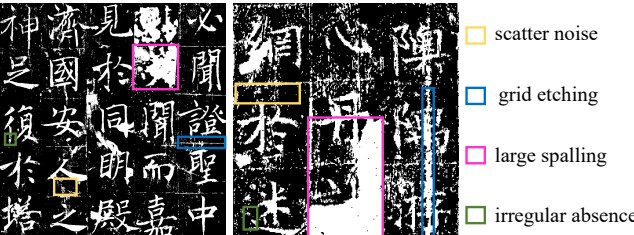

scatter noise

grid etching

large spalling

irregular absence

**Figure 3: Different degradation forms in the CIRI.**

principal criteria: (1) **Historical Diversity**. To accurately reflect the evolution and damage to inscriptions across different historical periods, we include a wide range of renowned inscription works spanning from the Han Dynasty (202 BC - 220 AD) to the Ming Dynasty (1368 AD - 1644 AD). (2) **Calligraphic Variety**. To encompass a diversity of calligraphy styles, our collection features works from the top ten ancient Chinese calligraphers, such as Zhenqing Yan, whose unique personal style evolved into the renowned Yan Style calligraphy, influencing the art form for millennia. (3) **Complex Degradations**. Focusing on the restoration task, we select rubbing images exhibiting a range of complex degradation patterns, thereby providing a realistic challenge for restoration efforts.

## 3.2 Data Processing, Annotation, and Synthesis

*3.2.1 Data Processing and Annotation.* The sources of existing rubbing images are diverse, affected by a range of factors such as photography, scanning, and mounting processes. These elements introduce a variety of complex backgrounds and lighting conditions into the images. However, *carefully restoring these background details does not significantly contribute to the primary goal of inscription restoration or facilitate the work of subsequent researchers.* To overcome this challenge and ensure a fair comparison, we implement a uniform binarization transformation across all rubbing images in our study. To enhance the diversity of our dataset, we also perform precise cropping and segmentation, resulting in a collection of 2000 real images. Moreover, each image has been manually annotated to detail its textual content and categorized according to its degradation level: easy, medium, or hard. It is critical to acknowledge that unified character-level region localization and annotation are not feasible for the real degradation inscription rubbing datasets. The diversity of degradation patterns often results in contiguous defect areas, obscuring the true boundaries of the characters.

*3.2.2 Degradation Analysis.* Our study delves into the various forms of degradation, as depicted in Figure 3. From both graphical and origination perspectives, degradation can be categorized into four distinct types: scatter noise, large spalling, grid etching, and irregular absence. For instance, "迷" in the second example lost glyph "辶". These degradation forms present two challenges. Firstly, the degradations introduce fuzziness, blurring the distinction between defects and original glyphs. Secondly, the covered radicals or characters inject uncertainty into text content understanding. The complexity and diversity of these degradations cannot be adequately mimicked using computer graphic methods. Consequently, we choose 1,000 images for extracting degradation areas. For each image, we manually identify 10 to 15 areas affected by degradation.

*3.2.3 Paired Data Synthesis.* Given the absence of intact images corresponding to real degraded rubbing instances, we devise a robust synthesis pipeline capable of generating image pairs closely resembling actual degradation conditions.

- **Data Preparation**. The foundation for our synthetic images is two-fold. On one hand, we select a variety of ancient Chinese inscriptions and scrolls. On the other hand, we opt for 105 traditional Chinese fonts and employ an efficient font imitation method [43] to facilitate the following generation.
- **Intact Image Synthesis**. Utilizing the aforementioned resources, we apply computer graphics methods [14] to render intact rubbing images. These images were diversified in attributes such as font, content, and layout.
- **Degradation Generation**. The process begins with the random combination of extracted real scatter noise, applied as a mask to each intact image. Subsequently, large spalling and synthetic convex hull [54] are integrated into the mask image, respecting the character area in each image to avoid overlap. The final steps includes the irregular erasure of some glyphs of characters and the addition of grid etching to certain images.

Through these elaborate steps, we successfully generated rubbing images, shown in Figure 4, that closely mimicd the real-life degradation scenarios encountered in ancient inscriptions.

During the collection process of existing real electronic rubbings, we find that complete inscriptions are frequently segmented into multiple rubbing image blocks. This segmentation facilitates detailed analysis and enhances the quality of the digital representations. Mirroring this approach in our data synthesis, we adopt a similar strategy, focusing on assembling rubbings into block-like configurations that encompass multiple rows and columns. This method not only aligns with conventional practices observed in real rubbings but also ensures that our synthesized data maintains a high degree of visual and structural integrity. Leveraging our flexible and controllable synthesis approach, we create 24,000 paired "intact-degraded", whose image size is set to 288 × 288, categorizing them according to the three levels of degradation in real rubbing samples.

Notably, To guarantee the quality of dataset construction, a strict double-round verification is conducted following the above phases. The entire manual procedure of data collection, annotation, synthesis, and verification demands around 1,000 person-hours.

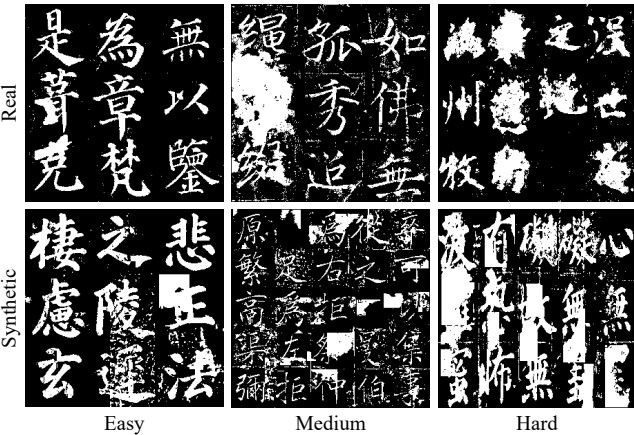

Figure 4: Some real and synthetic examples in the CIRI.

## 3.3 Dataset Statistics

Having synthetic rubbing images in hand, we strategically divide them into two subsets: 20,000 images are allocated as the training set, and the remaining 4,000 images constitute the synthesis testing set (abbr. Testing Set S). In parallel, for the real rubbing images, we select a portion of the degradation areas from 1,000 images to compile the Type I real testing set (abbr. Testing Set R-I). The residual 1,000 segments of real images, specifically those sections whose content and degradation areas were excluded from the synthesis of the training dataset, are assembled to form the Type II real testing set ((abbr. Testing Set R-II).

As depicted in Table 1, there exists a notable diversity in degradations and attributes of each subset.

## 4 METHODOLOGY

### 4.1 Overall Framework

The overall architecture of the proposed pipeline is depicted in Figure 5. For the input degraded rubbing image $r \in \mathbb{R}^{h \times w}$, the process begins with the GRM, which outputs a globally restored image $\tilde{x} \in \mathbb{R}^{h \times w}$. Following this, the TRM identifies and extracts the textual content $C = \{c^1, ..., c^n\}$ in $\tilde{x}$ and rectify it to $\tilde{C} = \{\tilde{c}^1, ..., \tilde{c}^n\}$ by retrieving the text corpus. Then, we convert this textual content into the corresponding images $S = \{s_{\tilde{c}}^1, ..., s_{\tilde{c}}^n\}$, proving the guidance of text content and character structure. In the final step, the LCM processes both $\tilde{x}$ and $S$ to produce the final intact rubbing image $\hat{x} \in \mathbb{R}^{h \times w}$. To maintain modularity and flexibility within our pipeline, each module is trained independently. As such, the network architecture of each functional module is designed to operate cohesively yet remain distinct, facilitating flexible improvements and modifications without impacting the integrity of the overall system.

### 4.2 Global Restoration Module

As mentioned above, a significant portion of the original rubbing data has invalid and unstable background information, prompting the adoption of a binary format for the rubbing images. This introduces a notable challenge in that the degraded and textual regions

become tangled, making it impossible to differentiate them based on pixel value distribution alone. Existing diffusion-based models [42, 54] conduct restoration within the continuous feature space, resulting in loss of information. In response, our global restoration module adopts the novel discrete diffusion paradigm [47], which is implemented by attention-based U-Net [28] with three pairs of symmetrical residual blocks.

*4.2.1 Training Procedure.* Unlike the original Denoising Diffusion Probabilistic Models (DDPM) [16], the transition process within our diffusion module cannot obey the Gaussian distribution, due to the binary characteristics of pixels in all inscription rubbing images. For instance, considering the intact rubbing image $x_{gt}$ during the training phase, each pixel value $x_{gt}[i]$ is constrained to the binary set $\{0, 1\}$.

In the noise addition phase, given $x_{gt}$ as $x_0$, we successively add Bernoulli distributions noise $\epsilon$ based on the time step $t$, as follows:

$$q(x^t|x^{t-1}) = \mathcal{B}(x^t; \alpha^t x^{t-1}, (1 - \alpha^t)I), \quad (1)$$

where $\alpha^t$ is a hyper-parameter ranging from 0 to 1. $\mathcal{B}$ denotes the Bernoulli distribution with parameter scale $p \in \{0, 1\}$. Leveraging the reparameterization trick [16], the progression can be generalized as:

$$x^t = \sqrt{\bar{\alpha}^t} x^0 + \sqrt{1 - \bar{\alpha}^t} \epsilon, \quad (2)$$

where $\epsilon \sim \mathcal{B}$ and $\bar{\alpha}^t = \prod_{j=0}^{t} \alpha_j \in [0, 1]$.

During the denoising phase, this module receives the degraded inscription rubbing image $r$ as conditions, formulated as:

$$p_{f_\theta}(x^{t-1}|x^t, r) = q(x^{t-1}|x^t, f_\theta(x^t, r, t)), \quad (3)$$

where $f_\theta$ represents the neural network with parameter $\theta$. Considering that the original output space of $f_\theta$ is consistent [47], this module adopts the threshold binarization operation to estimate the intact rubbing images directly.

The total process is supervised by the KL divergence loss, as:

$$\mathcal{L}_{gre} = \text{KL}(x^0, f_\theta(x^t, r, t)). \quad (4)$$

*4.2.2 Inference Procedure.* Having the pre-trained $f_\theta$, our module generates the restoration result $\tilde{x}$ after $T$ steps during the inference procedure, where each step can be written as:

$$x^{\tau-1} = f_\theta(x^\tau, r, \tau). \quad (5)$$

### 4.3 Text Rectification Module

While the GRM successfully addresses the majority of degradation issues in inscription rubbing images, two significant challenges persist within $\tilde{x}$. Firstly, some defects, such as peelings, which adhesion to or complete coverage over original strokes, lead the GRM to mistakenly erase these areas, resulting in incomplete strokes. Secondly, the capacity of GRM to the composition of legal characters relies heavily on its implicit learning from high-quality rubbing images. This reliance introduces a notable performance bottleneck when faced with the uncertainty of character forms due to complex defects, often leading to the generation of illegal character glyphs. To overcome these issues, we introduce a Text Rectification Module (TRM) that incorporates two types of external knowledge: text content and character structure, aiming to restore accurate and plausible glyphs for each character.

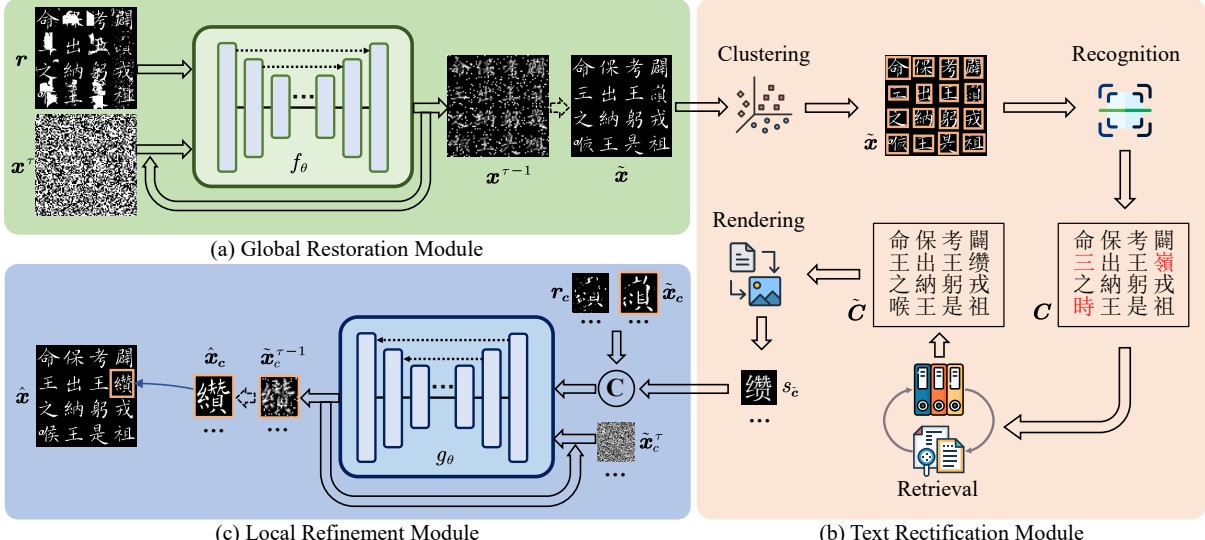

Figure 5: Illustration of the proposed pipeline. the ⓒ in subfigure (c) denotes the concatenate operation, and each dotted arrow in subfigure (a) and (c) denotes the final output of the diffusion-based inference procedure.

Given the global restoration output $\tilde{x} \in \mathbb{R}^{h \times w}$, we deploy a clustering algorithm to identify the layout of the inscription images, i.e., the location of each potential character $c$. By strategically padding the minimum bounding rectangle around each clustered region, we obtain the precise coordinates $b_c$. Thanks to the effective response of the GRM to large-sized degradation forms, this clustering algorithm achieves robust detection capabilities without additional training (refer to the ablation study for further details). Crucially, we employ a simple yet effective pre-trained ResNet-50 [15] as the backbone of the character recognizer, executing character-by-character recognition based on location $b_c$.

Subsequently, the central coordinates of each identified location are organized according to the traditional reading sequence of ancient Chinese texts, right-to-left and top-to-bottom, assembling these characters into a segment $\tilde{C}$. This segment is then cross-referenced against a corpus of ancient inscription texts to derive the accurate sentence $\tilde{C}$. Subsequently, this segment is transformed into an image format $S$, with each element directly corresponding to an individual character. This provides explicit guidance both on text content and the structure of the Chinese ideograph for each corresponding character block in the rubbing images. This rendering technique [14] has been validated for its efficacy within the text image community [6, 55].

The integration of external textual knowledge proves essential, particularly in the pursuit of achieving high-fidelity restoration of calligraphic styles. The correction of missing characters using references from other ancient manuscripts and intact inscriptions holds significant value in digital art preservation and aligns with the practices of restoration experts.

## 4.4 Local Refinement Module

Equipped with $S = \{s_{\tilde{c}}^1, ..., s_{\tilde{c}}^n\}$ and $\tilde{x}$, the Local Refinement Module (LRM) performs character-by-character restoration. For seamless integration with the rest of our framework, we adopt the diffusion-based restoration architecture utilized by the GRM, ensuring consistency across the entire restoration process.

*4.4.1 Training Procedure.* In the LRM, we leverage each rectified visual character $s_{\tilde{c}}$ along with its corresponding character image block $\tilde{x}_c$ as inputs for the diffusion training procedure. A key aspect of LRM is the implementation of a balanced training strategy designed to reduce biases towards the textural structure of standard characters, which might otherwise lead to inaccurate corrections throughout the restoration process. This involves mixing erroneous character image blocks produced by the GRM with correct character image blocks in a 1:1 ratio. Meanwhile, we add the original degraded image block $r_c$ to maintain the fidelity. Given the prior discussion of the detailed training procedure in § 4.2, we only present the loss function, as:

$$\mathcal{L}_{lre} = \text{KL}(x_c^0, f_\theta(\tilde{x}_c, s_{\tilde{c}}, r_c, t)), \tag{6}$$

where $x_c^0$ denotes the ground-truth character image block $x_c$.

*4.4.2 Inference Procedure.* To go beyond mere recognition accuracy and ensure the fidelity of character style, we perform the refinement to all original characters. For each character $c$, the binary restoration module $g_\theta$ is utilized to refine the image block $\tilde{x}_c$ based on its specific location coordinates $b_c$. Leveraging the rendered standard character image block $s_{\tilde{c}}$, this inference procedure at a given step can be simply written as:

$$\tilde{x}_c^{\tau-1} = g_\theta(\tilde{x}_c, s_{\tilde{c}}, r_c, \tau). \tag{7}$$

Then, each refined image block $\hat{x}_c$ is repositioned within the larger image according to its coordinates, producing the final restoration output $\hat{x}$.

**Table 2: The comparison results of visual presentation and textual integrity on Testing Set S. The bold denotes the best results. The †denotes these methods need external degradation masks for restoration.**

| Metric | Quality | | Recognition | | | End-to-End |
|---|---|---|---|---|---|---|
| Method | PSNR ↑ | SSIM ↑ | Top-1 Acc. ↑ | Top-5 Acc. ↑ | Macro Acc. ↑ | 1-NED ↑ |
| Degraded | 11.58 | 0.7430 | 0.3591 | 0.4586 | 0.1604 | 0.2514 |
| GDP [11]† | 14.04 | 0.8384 | 0.5441 | 0.6376 | 0.2494 | 0.4502 |
| CoPaint [48]† | 18.92 | 0.9006 | 0.8035 | 0.8816 | 0.4468 | 0.6538 |
| TSINIT [37] | 17.57 | 0.9186 | 0.8186 | 0.8868 | 0.4429 | 0.7244 |
| GSDM [54] | 20.15 | 0.9487 | 0.8952 | 0.9406 | 0.5329 | 0.8091 |
| Charformer [30] | 19.77 | 0.9438 | 0.9167 | 0.9555 | 0.5558 | 0.8342 |
| DocDiff [42] | 20.60 | **0.9532** | 0.9275 | 0.9629 | 0.5702 | 0.8452 |
| Ours | **21.16** | 0.9528 | **0.9692** | **0.9833** | **0.6443** | **0.8842** |
| Intact | - | - | 0.9971 | 0.9996 | 0.7064 | 0.9120 |

**Table 3: The end-to-end spotting results (1-NED ↑) on Testing Set R-I & R-II. The bold denotes the best results.**

| Dataset | Testing Set R-I | | | | Testing Set R-II | | | |
|---|---|---|---|---|---|---|---|---|
| Subset | Easy | Medium | Hard | Avg | Easy | Medium | Hard | Avg |
| Degraded | 0.2423 | 0.1483 | 0.0454 | 0.1823 | 0.3051 | 0.0795 | 0.0322 | 0.1644 |
| TSINIT [37] | **0.5937** | 0.5126 | 0.3417 | 0.5294 | 0.6084 | 0.4735 | 0.2695 | 0.4808 |
| GSDM [54] | 0.5909 | 0.5018 | 0.3393 | 0.5241 | 0.6040 | 0.4724 | 0.2776 | 0.4810 |
| Charformer [30] | 0.5789 | 0.4984 | 0.3606 | 0.5199 | 0.6047 | 0.4660 | 0.2548 | 0.4732 |
| Docdiff [42] | 0.5775 | 0.4888 | 0.3381 | 0.5127 | 0.5878 | 0.4571 | 0.2754 | 0.4681 |
| Ours | 0.5919 | **0.5365** | **0.4517** | **0.5528** | **0.6229** | **0.5314** | **0.3877** | **0.5351** |

## 5 EXPERIMENTS

### 5.1 Evaluation Protocal

The assessment of restoration outcomes on inscription rubbing datasets evaluates both visual presentation and textual integrity. To facilitate a comprehensive evaluation, we utilize a variety of metrics tailored to these aspects. (1)**Visual Presentation**: our main focus here is on assessing image quality and style consistency. For this purpose, we employ established metrics, including the Peak Signal-to-Noise Ratio (PSNR) in decibels (dB) and the Structural SIMilarity index (SSIM) [40]. These measures provide a quantitative foundation for assessing the restoration capacity, with a particular focus on maintaining clarity and stylistic fidelity. (2)**Textual Integrity**: Attention is given to layout accuracy and glyph fidelity in this aspect, referencing [31]. Here, we deploy a pre-trained YOLOv5 model [9] for precise character-level detection (with an IoU threshold of 0.5). Subsequently, a ResNet-50-based recognizer is employed to determine the recognition accuracy within each character region, leveraging metrics such as Top-1 Accuracy, Top-5 Accuracy, and Marco Accuracy [52]. The latter, particularly relevant for Ancient Chinese inscriptions, calculates accuracy for each category individually, thereby reflecting restoration performance in scenarios with long-tailed distributions. Furthermore, the Normalized Edit Distance (NED) is applied to comprehensively assess text integrity, measuring the similarity between the predicted text and its true counterpart for each image. This metric [51] can offer a refined measure of textual integrity.

It is important to note that a full evaluation of restoration models, encompassing both visual presentation and textual integrity, is feasible on the synthesis testing set. This is because only degraded rubbing images within this set possess corresponding intact counterparts and ground truth labels for character-level locations. Conversely, for real degraded rubbing images, the evaluation is restricted to the end-to-end spotting metric, NED, due to the absence of intact images and precise character-level ground truth. Meanwhile, we opt not to detail the performance of the detection on various restoration results in this study. This decision is driven by the high detection stability of YOLOv5, where the F1 scores for all results exceeded 0.98 (the outcomes of our pipeline surpass 0.99). Given these high scores, a comparison at finer granularity is deemed unnecessary.

### 5.2 Comparison with State-of-the-Art Methods

In our comparative analysis, we benchmark our proposed pipeline against a diverse set of prominent open-source methods.These include Restormer [45], CoPaint [48] and GDP [11] from natural image restoration, TSINIT [37] and GSDM [54] specializing in word-level text image inpainting, Charformer [30] and DocDiff [42] from document enhancement. For each method, we follow the default training settings and utilize the official implementations as applied to the CIRI dataset. In inference, we adaptively binarized their results to match the dataset. It is noteworthy that some methods, such as TransCNN-HAE [53] for blind image inpainting and Inpaint Anything [44] for general image inpainting, are found to be unsuitable for the specific challenges of inscription restoration.

We conduct the quantitative experiments on all three testing sets for the visual presentation and textual integrity. Here, we adopt a widely-used approach to text analysis [21], utilizing the ground-truth character locations on the restored images to evaluate pure recognition accuracy. Meanwhile, the detected character blocks by YOLOv5 are processed by the recognizer to determine the end-to-end spotting accuracy.

The quantitative results, presented in Table 2 and Table 3, reveal the following insights: (1) Our pipeline demonstrates exceptional performance in maintaining textual integrity. It nearly matches the accuracy of intact images in Testing Set R and handles complex degraded data effectively (see Table 3). This emphasizes the benefits of visual and textual guidance from autonomous retrieval and rectification. (2) The diffusion-based natural image restoration models show performance bottlenecks on all metrics, even when utilizing degradation masks, which are not available in real scenarios. This

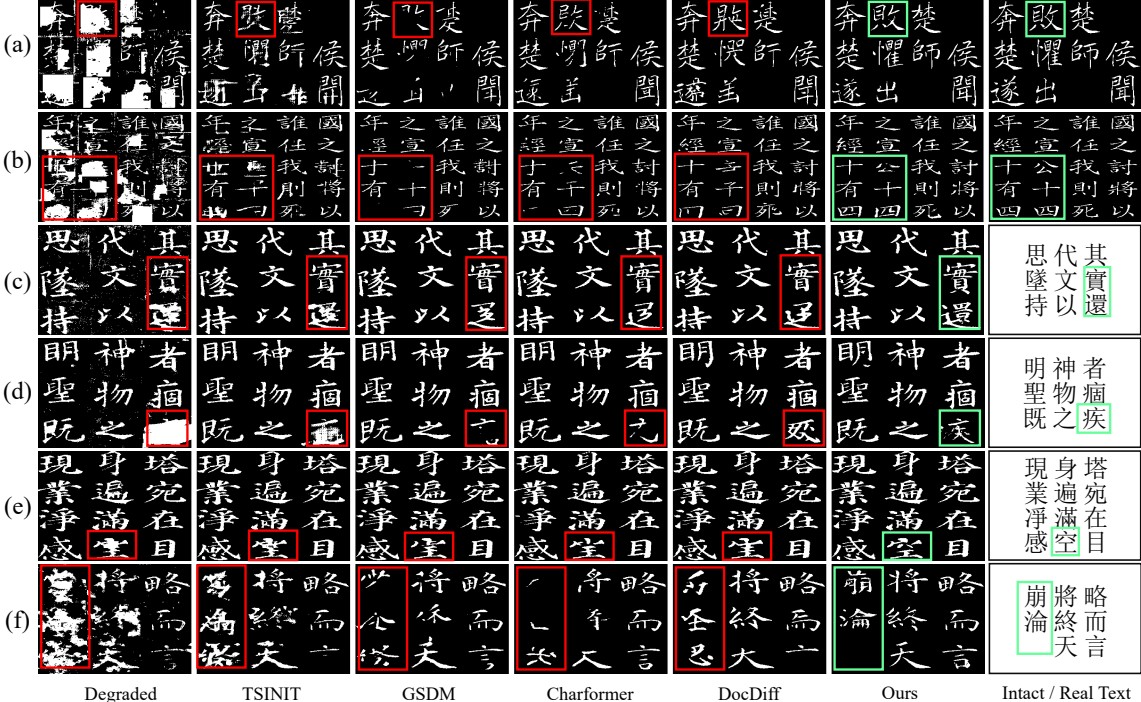

| Degraded | TSINIT | GSDM | Charformer | DocDiff | Ours | Intact / Real Text |

**Figure 6: Restoration results of different methods. (a)-(b) belong to Testing Set S, (c)-(d) belong to Testing Set R-I, and (c)-(d) belong to Testing Set R-II. The red rectangles and green rectangles denote the wrong and correct characters, respectively.**

indicates that significant domain differences exist between inscription and natural images, and targeted design is necessary. All these observations further demonstrate the efficiency of our pipeline.

It is worth noting that the improvement in image quality achieved through our restoration pipeline is relatively modest when compared to the enhancement in recognition rates. This discrepancy may be attributed to the sensitivity of PSNR and SSIM metrics, which focus on pixel-level information. Although the images restored by our pipeline exhibit a consistent and user-friendly style, they are constrained by the inherent complexities of the ill-posed problem in low-level vision tasks. Specifically, some strokes in the restored images show pixel-level positional deviations.

As shown in Figure 6, our pipeline excels at restoring the visual presentation reinforcing the discussions on quality metrics presented in our quantitative analysis. Here, we do not illustrate the outcomes from GDP [11] and CoPaint [48] for the consistency of different testing sets. First, leveraging integrated domain knowledge, our pipeline demonstrates exceptional capability in handling uncertain characters or radicals. For example, as depicted in Figure 6(a), it accurately restores the "貝" in the "敗", which is covered by peelings in the degraded image. Second, in more complex cases of degradation, our pipeline effectively identifies and reconstructs damaged Chinese characters with accurate glyphs, as shown in Figure 6(c) and (f). Lastly, the pipeline proves to be robust against complex and dense character configurations. For instance, Figure 6(a) highlights its ability to correctly restore the character "還" and avoid character misleading caused by defects, as mistaking "空" for "室" in Figure 6(f). Moreover, the various restoration models developed

using our synthesized dataset have shown differential restoration capabilities on real-world degraded inscription rubbings, demonstrating the generalizability of the CIRI dataset and the effectiveness of our synthesis approach.

## 6 CONCLUSION

In this paper, we present a systematic study of the restoration of inscriptions using artificial intelligence techniques. We introduce a pioneering Chinese Inscription Rubbing Image (CIRI) dataset, designed to explore both visual presentation and textual integrity in inscription restoration. By analyzing real collections, we develop a method to synthesize inscription rubbing images that mimic real-world degradation scenarios effectively. On top of it, we propose a multi-modal pipeline that employs a global-and-local strategy, seamlessly integrating knowledge of text and glyph structure to restore degraded inscription images. Comparative analysis with state-of-the-art restoration methods demonstrates the capability of our pipeline to handle diverse forms of degradation, highlighting the potential for liberating experts from time-consuming labor.

Looking ahead, the field of inscription restoration holds critical areas for further exploration. Our future research will focus on two main objectives: enhancing our restoration techniques to accommodate a broader range of calligraphy styles in an open-world corpus and addressing the challenges of restoring inscriptions with disordered content. These efforts are poised to make significant contributions to digital humanities, enabling archaeologists and epigraphers to perceive historical texts more authentically.

## ACKNOWLEDGMENTS

This work was supported by the National Natural Science Foundation of China (Nos. 62076062 and 62306070) and the Social Development Science and Technology Project of Jiangsu Province (No. BE2022811). Furthermore, the work was also supported by the Big Data Computing Center of Southeast University.

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
