# OpenReview forum: "Reproducing the Past: A Dataset for Benchmarking Inscription Restoration"
_acmmm.org/ACMMM/2024/Conference — MM2024 Oral_

### Official Review · Reviewer_VrpS · 2024-05-06

**Rating:** 4
**Confidence:** 3

**Summary:**

In this work, the authors introduced a multi-modal solution for the inscription restoration task. To this end, the Chinese Inscription Rubbing Image dataset was first constructed. They then proposed a restoration pipeline with global restoration, text rectification, and local refinement. Overall, I appreciate the dataset and the multi-modal solution.

**Strengths:**

The dataset is needed in the field of inscription restoration. And the experiments seems promising.

**Limitations:**

About the dataset:
1)Have these images been arranged according to their history, calligraphic style, and degradation? From current introduction, it seems that above arrangements are not available, resulting to limited application for this dataset. I also suggest the authors to comprehensively present experimental results according to above arrangements.
2)According to Fig. 5 and Sec 4, there should be ground-truth for text recognition and character restoration. How do the authors obtain these labels?

About the pipeline:
1)The proposed pipeline is practical but lacks novelty. What are the differences between global restoration module, local refinement module, and previous methods? It is also obvious that the text recognition relies on existing clustering and retrieval techniques. Please consider  and address this issue.
2)How these three modules are trained? How previous modules influence performance of the latter module, e.g., what will happen if the text rectification modules provides wrong results to local refinement module.
3)Please illustrate how easy, medium, and hard images are classified. The authors should provide necessary visualization in the manuscript.
4)Table 2 is confusing. How previous restoration methods such as CharFormer and DocDiff perform recognition? What does end-to-end mean?
5)Fig. 6 does not show results of recognition. More methods should be compared in tables and figures.

**Suitability:**

2

---

### Official Review · Reviewer_vePs · 2024-05-25

**Rating:** 4
**Confidence:** 2

**Summary:**

This paper constructs the Chinese Inscription Rubbing Image (CIRI) dataset, which includes a diverse collection of real inscription rubbing images. It also proposes a restoration framework that achieves visual consistency and textual integrity through global and local diffusion-based restoration processes and explicit incorporation of domain knowledge.

**Strengths:**

(1) The paper has a well-organized structure, and the description of details is relatively clear.
(2) The constructed dataset exhibits diverse styles and complex degradation forms.

**Limitations:**

(1) Experiments were conducted solely on the constructed dataset. It is suggested to add experiments on other benchmarks to compare and validate the generalizability of the proposed restoration framework.
(2) It is suggested to include the visualization analysis of intermediate results during the restoration process.
(3) It is recommended to perform the ablation analysis of the hyperparameters such as the alpha during the noise addition process to demonstrate the robustness of the proposed framework.

**Suitability:**

3

---

### Official Review · Reviewer_ffGu · 2024-05-25

**Rating:** 6
**Confidence:** 4

**Summary:**

This manuscript provides a inscription restoration dataset and an inscription restoration pipeline.

**Strengths:**

1. The dataset is rich, with clear focus. It unlocks the possibility of models for inscription restoration.
2. Using diffusion model for generating the inscription character is quite novel.
3. This paper is well-written.

**Limitations:**

The literature review on natural image restoration (Initial efforts) is limited.
Please cite BM3D and Non-local means.
@article{dabov2007image,
  title={Image denoising by sparse 3-D transform-domain collaborative filtering},
  author={Dabov, Kostadin and Foi, Alessandro and Katkovnik, Vladimir and Egiazarian, Karen},
  journal={IEEE Transactions on image processing},
  volume={16},
  number={8},
  pages={2080--2095},
  year={2007},
  publisher={IEEE}
}
and
@inproceedings{buades2005non,
  title={A non-local algorithm for image denoising},
  author={Buades, Antoni and Coll, Bartomeu and Morel, J-M},
  booktitle={2005 IEEE computer society conference on computer vision and pattern recognition (CVPR'05)},
  volume={2},
  pages={60--65},
  year={2005},
  organization={Ieee}
}.
Please also cite the paper on SSIM.
@article{wang2004image,
  title={Image quality assessment: from error visibility to structural similarity},
  author={Wang, Zhou and Bovik, Alan C and Sheikh, Hamid R and Simoncelli, Eero P},
  journal={IEEE transactions on image processing},
  volume={13},
  number={4},
  pages={600--612},
  year={2004},
  publisher={IEEE}
}

**Suitability:**

3

---

### Official Review · Reviewer_Ev61 · 2024-06-06

**Rating:** 5
**Confidence:** 2

**Summary:**

This paper presents a novel approach to restoring degraded ancient Chinese inscriptions. It introduces the Chinese Inscription Rubbing Image (CIRI) dataset, which includes both real and synthetically degraded images. The proposed multi-modal framework combines global and local restoration modules with domain knowledge to achieve high visual and textual fidelity. The study highlights the challenges of restoring inscriptions with complex degradations and demonstrates the effectiveness of the proposed methods through comprehensive evaluations, setting a new benchmark for future research in inscription restoration.

**Strengths:**

The paper's strengths include constructing the pioneering Chinese Inscription Rubbing Image (CIRI) dataset for ancient inscription restoration, proposing a robust multi-modal framework integrating global and local restoration techniques, and incorporating domain knowledge for high visual and textual fidelity. It provides comprehensive evaluations demonstrating significant improvements, is well-structured and clear, and has significant applications in archaeology and digital humanities. The dataset's diversity, effective synthesis approach, and modular framework design ensure scalability, flexibility, and authenticity in restored inscriptions, setting a new benchmark for future research and advancements in digital restoration technologies.

**Limitations:**

This paper has the following limitations:

1. While the CIRI dataset is comprehensive for Chinese inscriptions, the methods and findings may not directly apply to inscriptions from other  languages with different calligraphic styles and degradation patterns, i.e. oracle bone scripts. This limits the broader applicability of the proposed framework and dataset.

2. The synthetic generation of "intact-degraded" paired data, although designed to mirror real-world degradation, may not fully capture the complexity and variability of actual historical degradations. This reliance on synthetic data could affect the robustness and accuracy of the restoration models when applied to real-world scenarios.

3. The proposed multi-modal framework, which includes diffusion-based restoration and detailed text rectification processes, may involve high computational costs and require significant resources for training and implementation. This could limit the accessibility and practical use of the framework, particularly for institutions with limited computational resources.

**Suitability:**

3

---

### Meta-Review · Area_Chair_cPyC · 2024-07-01

**Recommendation:** Accept (Oral)
**Confidence:** 5

**Metareview:**

Four experts in the field reviewed this paper. This paper received 3 WA and 1 A in the final rating. Reviewers like the presentation and the rich dataset proposed. It greatly inspires research in inscription restoration. The rebuttal addressed the reviewer's concerns. Based on the scores, we recommend the acceptance of this paper.